# Effect of Single Nucleotide Polymorphisms in the Vitamin D Metabolic Pathway on Susceptibility to Non-Small-Cell Lung Cancer

**DOI:** 10.3390/nu14214668

**Published:** 2022-11-04

**Authors:** Laura Elena Pineda Lancheros, Susana Rojo Tolosa, José María Gálvez Navas, Fernando Martínez Martínez, Almudena Sánchez Martín, Alberto Jiménez Morales, Cristina Pérez Ramírez

**Affiliations:** 1Pharmacogenetics Unit, Pharmacy Service, University Hospital Virgen de las Nieves, 18014 Granada, Spain; 2Biomedical Research Center, Department of Biochemistry and Molecular Biology II, Institute of Nutrition and Food Technology “José Mataix”, University of Granada, Avda. del Conocimiento s/n., 18016 Granada, Spain; 3Department of Pharmacy and Pharmaceutical Technology, Social and Legal Assistance Pharmacy Section, Faculty of Pharmacy, University of Granada, 18071 Granada, Spain

**Keywords:** non-small-cell lung cancer, metabolic pathway of vitamin D, risk, biomarker, single-nucleotide polymorphisms, *CYP27B1*, *CYP2R1*, *GC*, *CYP24A1*, *VDR*

## Abstract

The pathogenesis of non-small-cell lung cancer (NSCLC) is complex, since many risk factors have been identified. Recent research indicates that polymorphisms in the metabolic pathway of vitamin D may be involved in both risk and survival of the disease. The objective of this study is to assess the effect of 13 genetic polymorphisms involved in the vitamin D metabolic pathway on the risk of suffering from NSCLC. We conducted an observational case-control study, which included 204 patients with NSCLC and 408 controls, of Caucasian origin, from southern Spain. The *CYP27B1* (rs4646536, rs3782130, rs703842, rs10877012), *CYP2R1* (rs10741657), *GC* (rs7041), *CYP24A1,* and *VDR* (BsmI, Cdx-2, FokI, ApaI, TaqI) gene polymorphisms were analyzed by real-time polymerase chain reaction. The logistic regression model, adjusted for smoking and family history of cancer, revealed that in the genotypic model, carriers of the *VDR* BsmI rs1544410-AA genotype were associated with a lower risk of developing NSCLC compared to the GG genotype (*p* = 0.0377; OR = 0.51; CI_95%_ = 0.27–0.95; AA vs. GG). This association was maintained in the recessive model (*p* = 0.0140). Haplotype analysis revealed that the AACATGG and GACATGG haplotypes for the rs1544410, rs7975232, rs731236, rs4646536, rs703842, rs3782130, and rs10877012 polymorphisms were associated with a lower risk of NSCLC (*p* = 0.015 and *p* = 0.044 respectively). The remaining polymorphisms showed no effect on susceptibility to NSCLC. The BsmI rs1544410 polymorphism was significantly associated with lower risk of NSCLC and could be of considerable value as a predictive biomarker of the disease.

## 1. Introduction

Lung cancer is one of the most serious malignancies, with the highest mortality (18% of cancer deaths worldwide) [1]. It also has a higher incidence in men (14.3%) and is the third most common cancer in women (8.4%), after breast and colorectal cancer. Taking both sexes together, it has the second highest incidence after breast cancer [1]. It is estimated that by 2040 the incidence of lung cancer will rise by 22.4% in Europe and that the increase in mortality will be 25.5% [2]. According to the latest cancer statistics, 236,740 new cases and 130,180 deaths are projected to occur in the United States in 2022 [3].

Lung cancer is a heterogeneous disease comprising various types. On this basis it can be classified primarily as small-cell lung cancer (SCLC, 13% of cases) and non-small-cell lung cancer (NSCLC, 83% of cases). The latter, in turn, can be divided into three subtypes: squamous cell carcinoma, adenocarcinoma, and large cell carcinoma [4,5,6]. Numerous risk factors for NSCLC have been discovered, including tobacco (which causes 80% of cases), exposure to radon, air pollution, previous history of lung disease, such as emphysema, bronchitis, asthma or chronic obstructive pulmonary disease (COPD), somatic mutations, and low serum levels of vitamin D [5,7].

Vitamin D (a secosteroid prohormone) has classically been considered to play a role in the regulation of calcium and phosphorus homeostasis, intervening in intestinal absorption, bone resorption, and reabsorption in the kidney [8]. Vitamin D has various functions in the body, in addition to those already mentioned, notably including its antiproliferative and proapoptotic ability, stimulating cell differentiation, acting as an antioxidant, and regulating the immune system, among other functions [8,9].

Several fat-soluble molecules are grouped together under the term vitamin D: vitamin D2 or ergocalciferol, vitamin D3 or cholecalciferol, and calcitriol or 1,25-dihydroxycalciferol (1,25(OH)2D), distinguished by their functions [7,10]. The first two are the inactive isoforms of the vitamin, whereas 1,25(OH)2D is its active form [11]. Cholecalciferol is obtained by the action of ultraviolet radiation (UVB 290–320 nm) on 7-dehydrocholesterol in the epidermis. Both cholecalciferol and ergocalciferol are ingested through the diet [8]. Both isoforms (vitamin D2 and D3) are transported in the bloodstream by being bound to VDBP (vitamin D binding protein), which carries the two precursors to the liver [12]. Here, the first hydroxylation takes place at position 25 through the action of the CYP2R1 or 25-hydroxylase enzyme. This reaction produces calcidiol or 25-hydroxyvitamin D. This metabolite remains in the blood for longer, so serum vitamin D levels are determined by calcidiol [8,11,12]. This is followed by a second hydroxylation of the molecule in the kidney, through the action of the CYP27B1 or α-1-hydroxylase enzyme, resulting in the active form of vitamin D: 1,25(OH)2D [7,10]. This metabolite is capable of binding to the vitamin D receptor (VDR), which is translocated to the nucleus to bind to the retinoid X receptor (RXR) and act as a transcription factor. Cells with high expression of VDR are sensitive to the antiproliferative activity of 1,25D3 [13]. Finally, vitamin D is degraded at mitochondrial level by the action of the CYP24A1 enzyme, preventing it from accumulating [11,12,14,15]. 

The genes that code for the enzymes involved in vitamin D metabolism (*GC*, *CYP2R1*, *CYP27B1*, *CYP24A1*, and *VDR*) are highly polymorphic [12,16]. Single nucleotide polymorphisms (SNPs) in these genes can alter both the functionality of the synthesized enzymes and the expression of each gene, and may therefore play an essential role in the onset, progression, and prognosis of NSCLC [12,17]. The influence of SNPs in these genes on the risk of developing lung cancer has been evaluated before, mainly in Asian populations [18,19,20,21,22,23,24,25,26,27,28,29,30,31]. However, there are few studies, and this highlights the need to carry out more research to assess the influence of these polymorphisms on the risk of suffering from NSCLC in various populations. Furthermore, no study has so far evaluated the combined effect of SNPs in *VDR*, *GC*, *CYP2R1*, *CYP27B1,* and *CYP24A1* on the risk of NSCLC. In addition, in a previous study we found a strong relationship between polymorphisms involved in the metabolic pathway of vitamin D and prognosis in patients with NSCLC [12], and we therefore believe that these SNPs may also be implicated in the pathogenesis of the disease. 

In the light of the foregoing considerations, this case-control study was designed with the aim of assessing the effect of polymorphisms in genes involved in vitamin D metabolism on the risk of suffering from NSCLC in Caucasian patients, specifically in southern Spain. 

## 2. Materials and Methods

We conducted an observational case-control study.

### 2.1. Study Population

This study involved 204 cases of NSCLC and 408 controls of Caucasian origin from southern Spain. The inclusion criteria for the cases were age 18 years or over, confirmed histological or cytological diagnosis of NSCLC (stages I–IV), and available clinical data. The controls were individuals aged over 18 years with no personal history of malignancies who had lived in the same geographical area and were recruited from the same hospital.

This case-control study was carried out in accordance with the Declaration of Helsinki and was approved by the Ethics and Research Committee of the Andalusian Public Health Service’s Biobank (Code: 1322-N-20). The subjects signed a written informed consent form for collection of blood or saliva samples and their donation to the Biobank. The samples were coded and treated confidentially.

### 2.2. Sociodemographic and Clinical Variables

The sociodemographic data include gender, age at diagnosis, smoking status, drinking status, family history of cancer, and previous lung disease. Individuals were classified as non-smokers if they had never smoked or had smoked fewer than 100 cigarettes in their lives, as ex-smokers if they had smoked 100 or more cigarettes in their lives but did not currently smoke, and as active smokers if they had smoked 100 or more cigarettes in their lives and currently smoke. Individuals were classified by standard drink units (SDUs) as non-drinkers if they were teetotalers or did not consume alcohol regularly, as active drinkers if their alcohol consumption was greater than 4 SDUs per day in men and greater than 2.5 SDUs per day in women, and as ex-drinkers if their alcohol consumption had been greater than 4 SDUs per day in men and greater than 2.5 SDUs per day in women but they did not currently drink [32]. Histopathologic data (tumor histology and stage) were also collected. The staging system used to classify the tumors was based on the guidelines of the American Joint Committee on Cancer [33]. 

### 2.3. Genetic Variables

#### 2.3.1. DNA Isolation

Blood samples (3 mL) were collected in BD Vacutainer^®^ K3E Plus blood collection tubes and saliva samples in BD Falcon™ 50 mL conical tubes (BD, Plymouth, UK). DNA was extracted using the QIAamp DNA Mini extraction kit (Qiagen GmbH, Hilden, Germany), according to the manufacturer’s instructions for purification of DNA from blood or saliva, and stored at −40 °C. The concentration and purity of the DNA were measured using the NanoDrop 2000™ UV spectrophotometer with 280/260 and 280/230 absorbance ratios. The DNA samples, isolated from blood or saliva, were preserved in the Biobank of the Hospital Universitario Virgen de las Nieves, part of the Andalusian Public Health Service’s Biobank.

#### 2.3.2. Detection of Gene Polymorphisms and Quality Control

We determined the gene polymorphisms by real-time PCR allelic discrimination assay using TaqMan^®^ probes (ABI Applied Biosystems, QuantStudio 3 Real-Time PCR System, 96 wells), following the manufacturer’s instructions (Table 1). Ten per cent of the results were confirmed by Sanger sequencing. Real-time PCR and Sanger sequencing were performed in the Pharmacogenetics Unit of the Hospital Universitario Virgen de las Nieves. The criteria for SNPs quality control were: (1) missing genotype rate per SNP < 0.05; (2) minor allele frequency > 0.01; (3) *p* value > 0.05 in Hardy-Weinberg equilibrium test; (4) missing genotype rate between cases and control < 0.05.

### 2.4. Statistical Analysis

We matched cases and controls by age and gender using the 1:2 propensity score matching method [34]. Quantitative data were expressed as the results (±standard deviation) for variables with normal distribution and medians or percentiles (25 and 75) for variables with non-normal distribution. We used the Shapiro–Wilks test to verify normality.

We determined the Hardy–Weinberg equilibrium and haplotype frequency through the D’ and r2 coefficients. The bivariate association between risk of NSCLC and polymorphisms was evaluated for multiple models (genotypic, additive, allelic, dominant and recessive), using the Pearson chi-square and Fisher exact tests, and assessed with the odds ratio and corresponding 95% confidence interval (CI). We defined the models as follows: genotypic (DD vs. Dd vs. dd), allelic (D vs. d), dominant ((DD, Dd) vs. dd), recessive (DD vs. (Dd, dd)), and additive (dd = 0, Dd = 1, DD = 2), where D is the minor allele and d the major allele. We used the Bonferroni adjustment for multiple comparisons. Unconditional multiple logistic regression models (genotypic, dominant, and recessive) were considered to determine the influence of possible confounding variables on the risk of suffering from lung cancer. 

All the tests were bilateral, with a significance level of *p* < 0.05, and were estimated using PLINK and R 4.0.2 software [35,36]. We performed linkage disequilibrium with Haploview 4.2 and haplotype analysis with SNPStats [37,38].

## 3. Results

### 3.1. Patient Characteristics 

The study included a total of 204 cases of NSCLC and 408 controls, whose clinicopathological characteristics are described in Table 2. The group of cases consisted of 73.5% men (150/204) and 26.5% women (54/204); the mean age at diagnosis was 61.1 ± 10.7 years; 47.9% were smokers (96/204); 76.2% were classified as non-drinkers (128/168); 50.5% had no family history of cancer (103/204); 65.7% had no previous lung diseases (134/204). As regards the features of the disease itself, 63.1% of the cases had adenocarcinoma (125/198) and 36.9% had squamous cell carcinoma (73/198); 68.3% were classified as stage IIIB–IV (138/202); and 31.7% as stage I, II or IIIA (64/202). The control group was made up of 68.4% men (279/408) and 31.6% women (129/408); the median age was 64 (52,75) years; 44.9% were classified as non-smokers (178/396); 60.0% were regarded as non-drinkers (251/369); 93.6% had no family history of cancer (382/408), and 65.2% had had no previous lung diseases (266/408).

There were statistically significant differences between the cases and the controls with respect to smoking status (*p* < 0.001; OR = 8.88; CI95% = 5.42–14.9; current smokers vs. non-smokers and *p* < 0.001; OR = 3.43; CI95% = 2.14–5.63; ex-smokers vs. non-smokers) and family history of cancer (*p* < 0.001; OR = 15.2; CI95% = 9.55–25.2; yes vs. no). No statistically significant differences were observed between the two groups in gender (*p* = 0.1898), age (*p* = 0.1030), drinking status (*p* = 0.1392), or previous lung disease (*p* = 0.9044).

### 3.2. Genotype Distribution

All the genotype frequencies observed agreed with those expected according to the Hardy–Weinberg equilibrium model, with the sole exception of *VDR* BsmI rs1544410 in the control group (*p* = 0.0459) (Appendix A). We found no statistical differences from those described in the Iberian population for this variant (*VDR* BsmI rs1544410 A allele: 0.4383 vs. 0.4393; *p* = 0.9989) [39]. The linkage disequilibrium (LD’) and r2 values are shown in Appendix A. In particular, the pairs *VDR* rs731236/rs7975232 (r2 = 0.545082; D’ = 0.959), *VDR* rs731236/rs1544410 (r2 = 0.615164; D’ = 0.907), *CYP27B1* rs4646536/rs3782130 (r2 = 0.850829; D’ = 0.933), *CYP27B1* rs4646536/rs10877012 (r2 = 0.815743; D’ = 0.925), and *CYP27B1* rs3782130/rs10877012 (r2 = 0.783361; D’ = 0.901) were in strong linkage disequilibrium (Figure 1). All the polymorphisms show minor allele frequencies (MAFs) greater than 1% and none of them was excluded from the analysis (Appendix A). The haplotype frequency estimates are presented in Appendix A.

### 3.3. Influence of Genetic Polymorphisms on the Risk of NSCLC

We used the genotypic, additive, allelic, dominant, and recessive models to perform the bivariate analysis between the gene polymorphisms and the risk of suffering from NSCLC (Appendix A). A statistically significant association was observed in the following SNPs: *VDR* BsmI rs1544410, in the genotypic (*p* = 0.0020), additive (*p* = 0.0122), allelic (*p* = 0.0121), and recessive (*p* = 0.0006) models, *VDR* TaqI rs731236 in the genotypic (*p* = 0.0299), and recessive (*p* = 0.0124) models, and *CYP24A1* rs6068816 in the genotypic (*p* = 0.0292), additive (*p* = 0.0316), allelic (*p* = 0.0353), and recessive (*p* = 0.0194) models (Appendix A). However, after the adjustment by the Bonferroni method had been made, the only SNP that maintained a statistically significant association with the risk of developing NSCLC was *VDR* BsmI rs1544410 in the genotypic and recessive models (Appendix A). In the genotypic model, patients carrying the *VDR* BsmI rs1544410-AA genotype had a lower risk of NSCLC relative to the GG genotype (p_Bonferroni-adjusted_ = 0.0361; OR = 0.457; CI95% = 0.26–0.76; AA vs. GG). Moreover, in the recessive model it was observed that patients carrying the *VDR* BsmI rs1544410-AA exhibited a lower risk of NSCLC than those carrying the G allele (p_Bonferroni-adjusted_ = 0.0082; OR = 0.442; CI95% = 0.26–0.70; AA vs. G) (Table 3). The logistic regression model adjusted for smoking and family history of cancer revealed that in the genotypic model carriers of the *VDR* BsmI rs1544410-AA genotype were associated with a lower risk of developing NSCLC compared to the GG genotype (*p* = 0.0377; OR = 0.51; CI95% = 0.27–0.95; AA vs. GG). This association was maintained in the recessive model, where patients carrying the AA genotype showed a lower risk of suffering from NSCLC than those carrying the G allele (*p* = 0.0140; OR = 0.49; CI95% = 0.27–0.85; AA vs. G) (Table 4). The other SNPs analyzed showed no statistically significant association with developing NSCLC in any of the models studied (Appendix A). In performing the haplotype analysis, we took account of the polymorphisms that were in strong linkage disequilibrium (Appendix A). We found that the AACATGG and GACATGG haplotypes, for the rs1544410, rs7975232, rs731236, rs4646536, rs703842, rs3782130, and rs10877012 SNPs, were associated with a lower risk of NSCLC (*p* = 0.015; OR = 0.63; CI95% = 0.44–0.91 and *p* = 0.044; OR = 0.11; CI95% = 0.01–0.94 respectively) (Table 5). 

## 4. Discussion

The pathogenesis of NSCLC is complex, since numerous risk factors have been identified, such as smoking, exposure to radon, air pollution, history of previous lung disease, somatic mutations, and low serum levels of vitamin D [5,7]. Recent research indicates that polymorphisms involved in the metabolic pathway of vitamin D are related both to survival of the disease [12,17,40,41,42,43] and to its pathogenesis [18,19,20,21,22,23,24,25,26,27,28,29,30,31]. It is therefore important to explore the impact of these associations in different populations. In this case-control study we investigate the influence of 13 genetic polymorphisms in five genes involved in the metabolic pathway of vitamin D on susceptibility to NSCLC in a Caucasian population (Spain).

Most of the effects of vitamin D are mediated by its binding to VDR (also known as NR1I1), a ligand-dependent transcription factor belonging to the nuclear receptor (NR) superfamily [44,45,46]. The VDR receptor and its ligand regulate the genes involved in calcium metabolism, cell growth, antiproliferation, differentiation, apoptosis, and adaptive/innate immune responses [44,45,46]. The SNPs of the *VDR* gene have been extensively studied. However, those that have proved most significant are BsmI (rs1544410), TaqI (rs731236), ApaI (rs7975232), FokI (2228570), and Cdx-2 (11568820). One of the SNPs that has been studied most in relation to the risk of NSCLC is BsmI (rs1544410), which is located in the 3′ UTR region, on intron 8 of the *VDR* gene, and gives rise to a change from adenine to guanine [18,19,20,21,22,23,24,25,26]. The functional polymorphisms located in this region may affect the function of VDR by regulating mRNA stability and protein translation efficiency, influencing the effect of vitamin D on tumor inhibition [47]. Our results show that in both the genotypic and the recessive models, patients carrying the *VDR* BsmI rs1544410-AA genotype had a lower risk of developing NSCLC; the data were adjusted to take account of smoking status and family history of cancer (Table 4). These results are in line with those described in previous studies. Specifically, a recent multiethnic meta-analysis (China, Turkey, Poland, and Tunisia) comprising ten articles (3046 cases/2716 controls) showed that the *VDR* BsmI rs1544410-AA genotype reduced the risk of lung cancer (*p* = 0.05; OR = 0.63; CI95% = 0.40–0.99; I2 = 50%; p_heterogeneity_ = 0.05; AA vs. GG and *p* = 0.02; OR = 0.78; CI95% = 0.63–0.97; I2 = 42%; p_heterogeneity_ = 0.11; AA vs. G) [28]. Similarly, in the analysis by cancer subtypes, the *VDR* BsmI rs1544410-AA genotype significantly reduced the risk of developing NSCLC (*p* < 0.00001; OR = 0.41; CI95% = 0.29–0.58; I2 = 56%; p_heterogeneity_ = 0.06; A vs. G and *p* = 0.0004; OR = 0.22; CI95% = 0.10–0.51; I2 = 0%; p_heterogeneity_ = 0.76; AA vs. G) [28].

Furthermore, we initially found in our study that the T allele for the TaqI (rs731236) SNP in the *VDR* gene showed a higher risk of NSCLC than the CC genotype (*p* = 0.0129). However, this association was not maintained after the Bonferroni adjustment. Nevertheless, these results are in line with what has been reported in the literature. A recent multiethnic meta-analysis (China, Turkey, Tunisia, Poland) based on 6 articles (2204 cases/2369 controls) showed that the T allele was associated with a higher risk of lung cancer compared to the CC genotype, both in the allelic model and in the recessive model (*p* = 0.02; OR = 1.14, CI95% = 1.02–1.27; I2 = 42%; p_heterogeneity_ = 0.12; T vs. C and *p* = 0.02; OR = 1.18; CI95% = 1.02–1.37; I2 = 46%; p_heterogeneity_ = 0.10; T vs. CC respectively) [28]. In addition, the analysis of subgroups by ethnic origin indicated that Asians carrying the T allele or the TT genotype showed a high risk of developing lung cancer (*p* = 0.008; OR = 1.56; CI95% = 1.12–2.17; I2 = 0%; p_heterogeneity_ = 0.99; T vs. C and *p* = 0.009; OR = 1.60; CI95% = 1.12–2.28; I2 = 0%; p_heterogeneity_ = 0.98; TT vs. C). However, this association was not found in Caucasians (*p* = 0.20; OR = 1.14; CI95% = 0.93–1.40; I2 = 57%; p_heterogeneity_ = 0.10; T vs. C and *p* = 0.22; OR = 1.22; CI95% = 0.88–1.69; I2 = 66%; p_heterogeneity_ = 0.05; TT vs. C) [28].

With regard to ApaI (rs7975232) and FokI (2228570), we found no association in our study between these SNPs and susceptibility to NSCLC. These results are in accordance with those published previously. For the ApaI SNP, a multiethnic meta-analysis (China, Turkey, Tunisia, Poland), published recently, consisting of 10 articles (3609 cases/3099 controls), found no statistically significant association in any of the models analyzed ([*p* = 0.09; OR = 0.83; CI95% = 0.66–1.03; I2 = 88%; p_heterogeneity_ < 0.00001; A vs. C]; [*p* = 0.11; OR = 0.74; CI95% = 0.51–1.07; I2 = 82%; p_heterogeneity_ < 0.00001; AA vs. C]; [*p* = 0.95; OR = 1.01; CI95% = 0.76–1.34; I2 = 80%; p_heterogeneity_ < 0.00001; AC vs. CC]; [*p* = 0.34; OR = 0.88; CI95% = 0.68–1.14; I2 = 78%; p_heterogeneity_ < 0.00001; A vs. CC] and [*p* = 0.08; OR = 0.71; IC95% = 0.48–1.04; I2 = 88%; p_heterogeneity_ < 0.00001; AA vs. C]) [28]. As for the FokI SNP, a multiethnic meta-analysis (China, Turkey, Tunisia, Poland) based on 5 articles (1362 cases/1474 controls) also failed to find a significant association with the risk of lung cancer in any of the models analyzed ([*p* = 0.30; OR = 1.14; CI95% = 0.89–1.47; I2 = 70%; p_heterogeneity_ < 0.009; C vs. T]; [*p* = 0.45; OR = 1.21; CI95% = 0.74–1.98; I2 = 62%; p_heterogeneity_ = 0.03; CC vs. CT]; [*p* = 0.63; OR = 0.88; CI95% = 0.53–1.47; I2 = 70%; p_heterogeneity_ = 0.01; CT vs. TT]; [*p* = 0.96; OR = 1.01; CI95% = 0.62–1.66; I2 = 71%; p_heterogeneity_ = 0.008; C vs. TT] and [*p* = 0.07; OR = 1.33; CI95% = 0.98–1.80; I2 = 55%; p_heterogeneity_ = 0.06; CC vs. T)] [28]. Similarly, we did not find a statistically significant association between susceptibility to NSCLC and the Cdx-2 (11568820) polymorphism. However, a meta-analysis (Poland, China) composed of 2 articles (1266 cases/1365 controls) showed that the T allele was associated with lower risk of lung cancer in both the heterozygous and the dominant model (*p* = 0.05; OR = 0.81; CI95% = 0.66–1.00; I2 = 0%; p_heterogeneity_ = 0.77; TC vs. CC and *p* = 0.03; OR = 0.80; CI95% = 0.65–0.98; I2 = 0%; p_heterogeneity_ = 0.99; T vs. CC respectively) [28].

Recently, the effect of the rs6068816 and rs4809957 SNPs of the *CYP24A1* gene on the risk of NSCLC has been studied [24,27,29,30,31]. This gene is responsible for synthesizing the enzyme involved in degrading vitamin D and thereby preventing it from accumulating. The results obtained in our study relate the rs6068816-C allele to lower risk of developing NSCLC in the bivariate analysis. However, this significance was not maintained after the Bonferroni adjustment. Nevertheless, our results are in line with those presented in the literature. A meta-analysis consisting of two studies conducted in Asian populations (China) (1056 cases/1302 controls) showed that the rs6068816 SNP was significantly associated with the risk of suffering from lung cancer. In particular, the rs6068816-C allele showed a protective effect from the development of NSCLC (*p* = 0.031; OR = 0.88; CI95% = 0.78–0.99; I2 = 0%; p_heterogeneity_ = 0.667; C vs. T and *p* = 0.049; OR = 0.85; CI95% = 0.72–1.00; I2 = 0%; p_heterogeneity_ = 0.955; CC vs. T). However, the significance was not maintained after the Bonferroni adjustment [29]. As for the rs4809957 SNP, no significant association was found in our study in any of the models analyzed. However, a previous study comprising 603 cases and 661 controls of Asian descent (China) found that rs4809957 was associated with the risk of NSCLC [27]. In particular, subjects carrying the rs4809957-AA genotype had a higher risk of suffering from NSCLC (*p* < 0.001; OR = 2.71; CI95% = 1.66–4.41; AA vs. G) [27].

The rs4646536, rs3782130, rs10877012, and rs703842 SNPs in the *CYP27B1* gene, the only one capable of synthesizing the α-1-hydroxylase enzyme, which is responsible for the second hydroxylation in the vitamin D activation process, showed no statistically significant associations with the risk of NSCLC in our study. The effect of the rs4646536 and rs703842 SNPs on susceptibility to developing lung cancer has not so far been evaluated. However, their effect on other cancers (breast, colorectal, and prostate, among others) has been assessed, without any statistically significant association being found [48]. On the other hand, two studies evaluating the effect of the rs10877012 and rs3782130 SNPs on the risk of developing NSCLC have been conducted. A study with 426 cases/445 controls in an Asian population (China) found no statistically significant association for rs10877012 ([*p* = 0.37; OR = 1.10; CI95% = 0.88–1.35; GG vs. GT]; [*p* = 0.37; OR = 0.95; CI95% = 0.76–1.21; GG vs. TT]; [*p* = 0.37; OR = 1.01; CI95% = 0.75–1.32; GG vs. T]) or rs3782130 ([*p* = 0.15; OR = 0.82; CI95% = 0.76–1.45; CC vs. CG]; [*p* = 0.15; OR = 1.03; CI95% = 0.89–1.34; CC vs. GG]; [*p* = 0.15; OR = 0.94; CI95% = 0.78–1.38; CC vs. G]) in any of the models analyzed [24]. Similarly, another study with 603 cases/661 controls of Asian descent (China) found no statistically significant associations for the rs10877012 SNP with respect to the risk of NSCLC in any of the genetic models analyzed ([*p* = 0.331; OR = 0.87; CI95% = 0.67–1.13; TT vs. GT]; [*p* = 0.331; OR = 1.10; CI95% = 0.76–1.58; TT vs. GG]; [*p* = 0.331; OR = 0.92; CI95% = 0.72–1.17; TT vs. G]; [*p* = 0.331; OR = 1.19; CI95% = 0.85–1.66; GG vs. T]) [27]. By contrast, a statistically significant association was found for the rs3782130 SNP in the genotypic model (*p* = 0.022; OR = 0.82; CI95% = 0.63–1.06; CC vs. CG), (*p* = 0.022; OR = 1.42; CI95% = 0.94–2.14; CC vs. GG) and in the dominant model (*p* = 0.022; OR = 1.60; CI95% = 1.09–2.34; GG vs. C) [27].

With regard to the rs10741657 SNP of the *CYP2R1* gene, located at the p15.2 locus on chromosome 11, this gene is responsible for encoding the main hydroxylase that converts vitamin D in the liver into the intermediate metabolite 25(OH)D (first hydroxylation) [49]. Our study found no statistically significant association with risk of suffering from NSCLC. These results are in line with a previous study in an Asian population (China), in which 603 cases/661 controls were assessed without finding a statistically significant association in any of the genetic models analyzed ([*p* = 0.216; OR = 0.82; CI95% = 0.64–1.06; GG vs. AG]; [*p* = 0.216; OR = 0.86; CI95% = 0.59–1.25; GG vs. AA]; [*p* = 0.216; OR = 0.83; CI95% = 0.65–1.06; GG vs. A]; [*p* = 0.216; OR = 0.96; CI95% = 0.68–1.36; AA vs. G]) [27].

The rs7041 SNP is located in exon 11 in domain III of the *GC* gene. This gene synthesizes the vitamin D binding protein (VDBP), which has immunomodulatory functions, specifically in the lung, related to activation of macrophages and chemotaxis of neutrophils [50]. Consequently, SNPs in the *GC* gene may alter the function of the protein and increase the risk of disease [51]. Our results did not show a statistically significant association between the rs7041 SNP and susceptibility to NSCLC in any of the genetic models analyzed. These results are in line with a meta-analysis comprising 3 articles with Asian populations (China) (1142 cases/1219 controls), which showed that the rs7041 SNP was not associated with risk of lung cancer in any of the models analyzed ([*p* = 0.52; OR = 0.90; CI95% = 0.65–1.25; I2 = 83%; p_heterogeneity_ = 0.003; G vs. T]; [*p* = 0.17; OR = 0.65; CI95% = 0.36–1.19; I2 = 63%; p_heterogeneity_ = 0.07; GG vs. TT]; [*p* = 0.93; OR = 0.97; CI95% = 0.56–1.70; I2 = 89%; p_heterogeneity_ = 0.0001; GT vs. TT]; [*p* = 0.80; OR = 0.93; CI95% = 0.55–1.59; I2 = 88%; p_heterogeneity_ = 0.0002; G vs. TT]; [*p* = 0.12; OR = 0.67; CI95% = 0.40–1.11; I2 = 52%; p_heterogeneity_ = 0.12; GG vs. T]) [28]. However, when the subgroup analysis by cancer subtypes was performed, it showed that the G allele and the GG genotype might be protective factors against the development of NSCLC in an Asian population (China) (*p* = 0.004; OR = 0.76; CI95% = 0.62–0.92; I2 = 53%; p_heterogeneity_ = 0.14; G vs. T and *p* = 0.03; OR = 0.58; CI95% = 0.35–0.95; I2 = 51%; p_heterogeneity_ = 0.15; GG vs. T respectively) [28].

The main limitation of this study is the limited size of the sample compared to other studies, particularly with regard to cases. This may have prevented detection of associations of some polymorphisms. However, despite this limited sample size, after the Bonferroni adjustment was applied to avoid false-positive associations, the effect of *VDR* BsmI rs1544410 on susceptibility to NSCLC remained. The strengths of our study include a very homogeneous cohort of cases, consisting solely of patients with NSCLC diagnosed by the same team of pathologists and recruited in the same geographical area, increasing their uniformity.

To summarize, our results suggested that the BsmI rs1544410 polymorphism in the VDR gene may act substantially as a protective factor against developing NSCLC. Further studies in different populations could help to find additional associations between other genes and polymorphisms in the vitamin D metabolic pathway and the risk of NSCLC.

## 5. Conclusions

The *VDR* BsmI rs1544410 gene polymorphism was significantly associated with a lower risk of NSCLC. No influence on the risk of developing NSCLC was found in our patients for the following gene polymorphisms: *VDR* Cdx2 rs11568820, *VDR* TaqI rs731236, *VDR* ApaI rs7975232, *VDR* FokI rs2228570, *CYP27B1* rs4646535, *CYP27B1* rs3782130, *CYP27B1* rs10877012, *CYP27B1* rs703842, *CYP24A1* rs6068816, *CYP24A1* rs4809957, *CYP2R1* rs10741657, and *GC* rs7041. 

## Figures and Tables

**Figure 1 nutrients-14-04668-f001:**
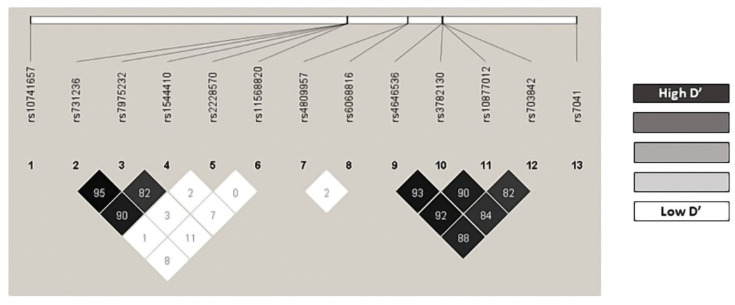
Linkage disequilibrium.

**Table 1 nutrients-14-04668-t001:** Gene polymorphisms and TaqMan^®^ ID.

Gene	Location, SNP	dbSNP ID	Assay ID
*VDR* (12q13.11)	Exon 8, G > A	rs1544410 (BsmI)	AN324M4 *
	Exon 1, G > A	rs11568820 (Cdx-2)	C___2880808_10
	Exon 2, C > T	rs2228570 (FokI)	C__12060045_20
	Exon 8, C > A	rs7975232 (ApaI)	C__28977635_10
	Exon 9, T > C	rs731236 (TaqI)	C___2404008_10
*CYP27B1* (12q14.1)	Intron 6, T > C	rs4646536	C__25623453_10
	Promoter 5′, G>A/G>C	rs3782130	ANGZRHH *
	5′ UTR, A > G 3′ UTR, A > G	rs10877012 rs703842	C__26237740_10 ANH6J3F *
*CYP24A1* (20q13.2)	Exon 6, G > A	rs6068816	C__25620091_20
	3′ UTR, G > C	rs4809957	C___3120981_20
*GC* (4q13.3)	Exon 11, T > G	rs7041	C___3133594_30
*CYP2R1* (11p15.2)	5′ UTR, A > G	rs10741657	C___2958430_10

* The SNPs were analyzed using custom assays by ThermoFisher Scientific (Waltham, MA, USA).

**Table 2 nutrients-14-04668-t002:** Clinico-pathologic characteristics of NSCLC cases and controls.

	Cases		Controls		χ^2^	*p*-Value	Reference	OR	CI95%
	N	n (%)	N	n (%)					
Gender	204		408						
Female		54 (26.5)		129 (31.6)	1.7189	0.1898			
Male		150 (73.5)		279 (68.4)					
Age	204	61.1 ± 10.7	408	64 (52,75)		0.1030 *			
Tobacco consumption	204		396						
Current-smokers		96 (47.96)		68 (17.2)	81.179	<0.001	Non-smokers	8.88	5.42–14.9
Former-smokers		81 (39.71)		150 (37.9)				3.43	2.14–5.63
Non-smokers		27 (13.24)		178 (44.9)				1	
Alcohol consumption	168		369						
Current-drinkers		34 (20.24)		104 (28.2)	3.9433	0.1392			
Former-drinkers		6 (3.57)		14 (3.8)					
Non-drinkers		128 (76.2)		251 (60.0)					
Family history of cancer	204		408						
Yes		101 (49.5)		26 (6.37)	160.35	<0.001	No	15.2	9.55–25.2
No		103 (50.5)		382 (93.6)					
Previous lung disease	204		408						
Yes		70 (34.3)		142 (34.8)	0.0144	0.9044			
No		134 (65.7)		266 (65.2)					
Histology	198								
Adenocarcinoma		125 (63.1)							
Squamous cell carcinoma		73 (36.87)							
Tumor stage	202								
I, II or IIIA		64 (31.7)							
IIIB or IV		138 (68.3)							

* *p*-value for t test; Shade means the value is significant. N means the whole number of patients considered; n means the number of patients in subgroups.

**Table 3 nutrients-14-04668-t003:** Influence of *VDR* BmsI (rs1544410) gene polymorphism on risk of NSCLC.

Models	Genotype	Cases [n (%)]	Controls [n (%)]	*p*-Value ^a^	Adjusted *p*-Value ^b^	OR ^c^	CI95%
Genotypic	GG	71 (34.8)	126 (31.2)	0.00278	0.0361	1	
	AG	108 (52.9)	181 (44.8)			1.058	0.72–1.54
	AA	25 (12.3)	97 (24.0)			0.457	0.26–0.76
Dominant	A	133 (65.2)	278 (68.8)	0.3684	1		
	GG	71 (34.8)	126 (31.2)				
Recessive	AA	25 (12.3)	97 (24.0)	0.00063	0.0082	0.442	0.26–0.70
	G	179 (87.7)	307 (76.0)				
Allelic	A	158 (38.7)	375 (46.4)	0.01076	0.1399		
	G	250 (61.3)	433 (53.6)				
Additive	-	-	-	0.01217	0.1582		

CI: confidence interval; OR: odds ratio. ^a^
*p*-value for χ2-test. ^b^
*p*-value for Bonferroni correction. ^c^ Unadjusted or crude ORs. Shade means the value is significant.

**Table 4 nutrients-14-04668-t004:** Influence of clinical characteristic and *VDR* BmsI (rs1544410) gene polymorphism on risk of NSCLC.

	Genotypic						Dominant			Recessive			Additive		
	AA vs. GG			AG vs. GG			A vs. GG			AA vs. G			A vs. G		
	*p*-Value	OR	CI95%	*p*-Value	OR	CI95%	*p*-Value	OR	CI95%	*p*-Value	OR	CI95%	*p*-Value	OR	CI95_%_
Tobacco consumption															
Current smokers	<0.001	6.11	3.51–10.8	<0.001	6.11	3.51–10.8	<0.001	5.91	3.40–10.4	<0.001	6.10	3.50–10.8	<0.001	5.96	3.43–10.5
Former smokers	0.0003	2.64	1.57–4.55	0.0003	2.64	1.57–4.55	0.0004	2.58	1.53–4.43	0.0003	2.64	1.56–4.54	0.0003	2.59	1.54–4.46
Family history of cancer															
Yes	<0.001	10.9	6.67–18.4	<0.001	10.9	6.67–18.4	<0.001	11.3	6.90–18.9	<0.001	10.9	6.68–18.4	<0.001	11.1	6.81–18.8
rs1544410	0.0377	0.51	0.27–0.95	0.7786	1.06	0.67–1.70	0.5478	0.87	0.56–1.36	0.0140	0.49	0.27–0.85	0.0752	0.76	0.57–1.02

Shade means the value is significant.

**Table 5 nutrients-14-04668-t005:** Haplotype association with risk of NSCLC.

	rs1544410	rs7975232	rs731236	rs4646536	rs703842	rs3782130	rs10877012	Freq	OR (CI95%)	*p*-Value
1	G	C	T	A	T	G	G	0.3283	1.00	---
2	A	A	C	A	T	G	G	0.2364	**0.63 (0.44–0.91)**	0.015
3	A	A	C	G	C	C	T	0.1053	1.00 (0.65–1.53)	0.99
4	G	A	T	A	T	G	G	0.082	0.95 (0.58–1.53)	0.82
5	G	C	T	G	C	C	T	0.0661	0.90 (0.48–1.65)	0.72
6	A	C	T	A	T	G	G	0.0241	0.00 (−Inf–Inf)	1
7	G	A	T	G	C	C	T	0.019	0.40 (0.09–1.84)	0.24
8	A	A	T	A	T	G	G	0.0136	1.26 (0.27–5.88)	0.77
9	G	A	C	A	T	G	G	0.0133	**0.11 (0.01–0.94)**	0.044
10	G	C	T	G	T	C	T	0.011	0.00 (−Inf–Inf)	1

Global haplotype association *p*-value: <0.0001. Inf: infinite; Freq: haplotype frequency; Shade means the value is significant.

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
