# Peer review of "Effect of Single Nucleotide Polymorphisms in the Vitamin D Metabolic Pathway on Susceptibility to Non-Small-Cell Lung Cancer"

_nutrients, 2022, doi:10.3390/nu14214668_

Round 1

Reviewer 1 Report

1. Please discuss the reason why patients with the VDR BsmI rs1544410-AG genotype showed a higher risk of NSCLC than carriers of the GG genotype, while those carrying the rs1544410-AA genotype had relatively the lower risk.

2. Please validate the values in Table 2 and the section “3.1. Patient characteristics” in Result.
e.g.
“25.6% women” should be 26.5% women.
“61.1 ± 10.72” 
should be shown as 61.1 ± 10.7.
The value in “no previous lung diseases (103/204)” should be 134/204.
It is unclear what “the median age was 64 [52,75] years” means.

3. In line 335: "1alpha-hidroxylase" should be corrected as shown in line 72.

Author Response

Answer:

Reviewer 1. Comment 1:

  1. Please discuss the reason why patients with the VDR BsmI rs1544410-AG genotype showed a higher risk of NSCLC than carriers of the GG genotype, while those carrying the rs1544410-AA genotype has relatively the lower risk.

Thank you very much for your positive and constructive comment. Apparently, AG vs. GG genotype is related to a higher risk of developing NSCLC. Otherwise, statistical analysis results are not significant due to the OR value (extremely close to 1) and IC95% value (which englobes 1). Thus, p-value from bivariant statistical analysis is annulled. According to what has been mentioned above, we have removed it from the text to avoid confusion.  

Reviewer 1. Comment 2:

  1. Please validate the values in Table 2 and the section “3.1. Patient characteristics” in Result.

e.g.

“25.6% women” should be 26.5% women.

“61.1 ± 10.72” should be shown as 61.1 ± 10.7.

The value in “no previous lung diseases (103/204)” should be 134/204.

It is unclear what “the median age was 64 [52,75] years” means.

In the Results section, in “3.1. Patient characteristics” precisely, we have validated the values in Table 2 as suggested.

Changes in the text; Page 4; Line 175 and 177

The study included a total of 204 cases of NSCLC and 408 controls, whose clinicopathological characteristics are described in Table 2. The group of cases consisted of 73.5% men (150/204) and 26.5% women (54/204); the mean age at diagnosis was 61.1 ± 10.7 years; 47.9% were smokers (96/204); 76.2% were classified as non-drinkers (128/168); 50.5% had no family history of cancer (103/204); 65.7% had had no previous lung diseases (134/204).

Changes in the text; Page 5; Table 2

26.5

According to the explanation of the median age meaning, after Shapiro-Wilks normality test application those non-normal distribution quantitative variables were expressed as follows: percentile 50 [percentile 25, percentile 75]. Thus, 64 is percentile 50 of age which means it is the median, while [52,75] are percentile 25 and 75 respectively.

Reviewer 1. Comment 3:

  1. In line 335: “1alpha-hidroxylase” should be corrected as shown in line 72.

In the discussion section we have done the correction as suggested.

Changes in the text; Page 9; Line 335

“α-1-hydroxylase”

Reviewer 2 Report

The pathologic pathway of lung cancer is not fully elucidated. Therefore data concerning the Vitamin D metabolic pathway which may predict susceptibility to the development of lung cancer are important.  Moreover, this case-control study is well-designed and the results are appropriately presented. The sample size is adequate with 204 patients with lung cancer and 400 controls. Data retained significance at logistic regression analysis and the recessive test.

Author Response

Authors are enormously grateful for your positive and constructive comment.

Reviewer 3 Report

Altogether the study appear to be scientifically sound. Maybe the authors could detail a little more about how cases and controls were matched. Time elapse etc. Unfortunately cases and controls were not matched for the major lung cancer risk factors: smoking status.

Author Response

Altogether the study appears to be scientifically sound. Maybe the authors could detail a little more about how cases and controls were matched. Time elapse etc. Unfortunately cases and controls were matched for the major lung cancer risk factor: smoking status.

Thank you very much for your comment, we see your point. We tried to perform case and controls match according to such an important variable as smoking. However, it was not feasible due to the number of cases. Otherwise, those variables (smoking and familiar history of cancer) were included in multivariant statistical analysis. We will take account of your comment for further studies.
